# Induced Aggregation of Epoxy Polysiloxane Grafted Gelatin by Organic Solvent and Green Application

**DOI:** 10.3390/molecules24122264

**Published:** 2019-06-18

**Authors:** Zhen Zhang, Dongmei Zhang, Huijun Ma, Jing Xu, Tianduo Li, Zhaoning Cai, Haifeng Chen, Jinghui Zhang, Hao Dong

**Affiliations:** 1Shandong Provincial Key Laboratory of Molecular Engineering, School of Chemistry and Pharmaceutical Engineering, Qilu University of Technology (Shandong Academy of Sciences), Jinan 250353, China; zhangzhen950305@163.com (Z.Z.); 17862182055@163.com (H.M.); litianduo@163.com (T.L.); caizhaoning1221@163.com (Z.C.); chenhaifeng529@163.com (H.C.); zjh804300441@163.com (J.Z.); d237658934@163.com (H.D.); 2Shandong Institute for Food and Drug Control, Jinan 250101, China; zhangdm1000@163.com

**Keywords:** amphiphilic polymers, organic solvent, aggregation, coating, sewage treatment

## Abstract

In this paper, we studied the aggregation of amphiphilic polymer epoxy-terminated polydimethylsiloxane (PDMS-E) grafted gelatin (PGG) in water induced by methanol, ethanol, 2-propanol, acetone, tetrahydrofuran (THF), and 1,4-dioxane. The aggregation pattern of the polymer was monitored by infrared spectroscopy, X-ray diffraction, transmission electron microscopy, and scanning electron microscopy. It was revealed that the aggregate morphology showed clear dependence on the solvent polarity. The PGG aggregates had regular spherical morphology in polar solvents, including water, methanol, ethanol, 2-propanol, and acetone. The coating performance was evaluated by X-ray photoelectron spectroscopy and friction experiment, and PGG and acetone coating exhibited excellent coating performance on the surface of pigskin. Gel was formed in acetone and tetrahydrofuran (THF) with the slow evaporation of solvent, and this property can possibly be applied to industrial sewage treatment. White precipitate and soft film were formed in non-polar 1,4-dioxane.

## 1. Introduction

The aggregation of protein/peptide–polymer conjugates in a selective solvent has attracted increasing attention due to their capacity for the formation of a variety of hierarchical nano- and micro-structures or gel [1], which can be applied to biosensors [2], functional materials [3], bioimaging [4], tissue engineering [5] or drug delivery [6,7,8,9]. The aggregation of the conjugates is a synergistic effect of intermolecular noncovalent interactions, including hydrogen bonding, π-π, electrostatic, hydrophobic, and van der Waals interactions [10]. The synergistic effect can be disturbed by solvent polarity or solvent selectivity [11,12,13,14].

The polarity of solvents can influence the power of hydrogen bonding interaction between solute molecules and solvents [15]. Hydrogen bonding between biomolecules and solvents are considered as a dominant factor among the synergistic effect [16]. Even tiny amounts of solvents (e.g., water) will disturb the synergistic effect of the intermolecular noncovalent interactions and further change the aggregation of the conjugates [10]. Sahnawaz Ahmed report that the peptide-perylenediimides conjugate formed fiber-like morphology in relatively non-polar solvents (THF and CHCl_3_), while in more polar solvents (MeOH, acetone) spherical morphology could be found. Their result showed that the aggregations of the peptide-perylenediimides conjugate clearly depend on the solvent polarity. In polar solvents, the conjugate aggregates more efficiently than in the non-polar solvents, and with decrease in solvent polarity, the dimension of the nano-structures increased [17]. Matthews and Gil report that crystallization of peptides can be tuned by ethanol or methanol for preparation of biological materials [18,19].

The polarity of a solvent molecule is related to its dielectric constant (Ɛ). Normally, when Ɛ is lower than 15, the solvent is considered nonpolar. When Ɛ ranges from 15 to 25, the name medium polar solvent is given. When Ɛ is larger than 25, it is a strong polar solvent. In the paper, solvents, including strong polar water, methanol, and ethanol, medium polar solvents 2-propanol and acetone, and nonpolar solvents tetrahydrofuran and 1,4-dioxane, are selected to induce the aggregation of mono epoxy-terminated polydimethylsiloxane (PDMS-E) grafted gelatin (PGG) that was synthesized in our previous work [20]. Alcohol, which is thought of as a protic solvent, can be used as both the hydrogen donor and acceptor. Acetone, 1,4-dioxane, and tetrahydrofuran, which are aprotic solvents but contain highly electronegative oxygen, are more likely to accept hydrogen, being that they are hydrogen acceptors [21]. The differing of hydrogen bonding ability in a series of solvents of varying polarity can dramatically affect morphologies, sizes, and functions of aggregates [22,23,24,25,26].

In addition, solvent selectivity is an essential factor for disturbing aggregation patterns of amphiphilic block copolymers with covalently bonded but distinctive blocks. For example, under selectivity solvent mediation, amphoteric block polymer as a material can form various shapes of asymmetric nanoparticles [27,28], nanorings [29], hollow spheres, and so on [30]. Gelatin, as a renewable, cheap, and water soluble protein material, has been widely used in packaging and pharmaceutical and medical applications [31]. PDMS is an inorganic polymer exhibiting excellent performances, such as non-toxicity, air permeability, pliability, low glass transition temperature, is stable and easy to disperse, and it is widely used in the chemical or biological industries [32,33,34]. The compatibility between gelatin and PDMS in selective solvent plays an important role in inducing aggregation of PGG, and following this the diverse aggregation patterns that emerge in a series solvent of varying polarity.

In this study, infrared spectroscopy, X-ray diffraction, transmission electron microscopy, and scanning electron microscopy were used to characterize the aggregation pattern of PGG. The regular spherical morphology formed in polar solvents, including water, methanol, ethanol, and acetone, which presented a desired coating on the pigskin with excellent wear-resistance and radiation resistance. Gel was formed in acetone and THF with the slow evaporation of solvent, and this property can possibly be used for industrial sewage treatment.

## 2. Results and Discussion

### 2.1. Chemical Modification of Gelatin by PDMS-E

The chemical modification of gelatin through grafting reactions between free −NH_2_ groups and PDMS-E was performed in water. We can see in Figure 1 that grafting density is significantly affected by surfactant structure and concentration. Also, the conversion degree of free −NH_2_ groups reached the first peak values (24.77%) at 3.0 g L^−1^. The results show that the variation of the conversion degree with the increase of concentration is not monotonic. It can be deduced that the structure and concentration of surfactants has a large effect on grafting density. Grafting density is enhanced by the compatibility of the two phases and affects aggregation of product in the select solvent. In the work, the reaction condition (SDS, 3.0 g L^−1^) is chosen, which brings the highest grafting density.

### 2.2. Aggregation of PGG Induced by Solvents

In this study, organic solvents, including methanol, ethanol, isopropanol alcohol, acetone, THF, and 1,4-dioxane, were selected to tune the aggregation of PGG in water. From the data of dielectric constant, it can be inferred that the order in which polarity decreases is: water > methanol > ethanol > isopropanol alcohol > acetone > THF > 1,4-dioxane. In addition, water, methanol, ethanol, and isopropanol alcohol are protic solvents that can be both hydrogen donor and acceptor. Acetone, THF, and 1,4-dioxane, which contains highly electronegative oxygen atoms, can be regarded as hydrogen acceptors [21]. The difference of hydrogen bonding ability for these solvents induces diverse aggregation. In addition, the selectivity of gelatin component and PDMS component in solvent can lead to a spontaneous aggregation.

Figure 2 shows that PGG presented different morphologies in different solvents. Spherical aggregates were formed in methanol, ethanol, isopropanol alcohol, and acetone system at a PGG solution/solvent ratio of 1:1 (*v/v*; Figure 2a–d), and smaller-scale spherical aggregates were observed as the PGG solution/solvent ratio decreased to 1:2 (*v/v*; Figure 2g–i). However, in the acetone system, the decreasing PGG solution/acetone ratio resulted in the formation of coacervate (Figure 2d versus Figure 2j). Interestingly, a double-layer structure was formed in the THF-water system at a PGG solution/solvent ratio of 1:1 (*v/v*) (Figure 2e). The structure evolved to spherical aggregate at a PGG solution/THF ratio of 1:2 (*v/v*, Figure 2k). Disordered aggregates were observed in the 1,4-dioxane system (Figure 2f,l). These results suggest that the morphology transformations of PGG aggregates closely depend on the solvent polarity and the hydrogen bonding ability. In polar protic solvent, including methanol, ethanol, and isopropanol alcohol, spherical aggregates tend to form. With the increasing ratio of solvent, the scale of aggregate decreases. However, in nonpolar aprotic solvents THF and 1,4-dioxane, complex morphologies can be obtained, such as double-layer, disordered aggregates or coacervates.

Mirsky and Pauling were the first to suggest that hydrogen bonding was the dominant force of protein folding [35]. This folding force becomes more favorable as the number of polar groups increases [36]. This means that hydrogen bonding, which is affected by solvent polarity, plays a prominent role in controlling the conformation of the protein component. In general, the FTIR spectra of the different samples showed similar peaks and bands of amide A [37], amide I, amide II, and amide III, although with slight differences in the location of the peaks. IR spectroscopy is one of the powerful tools for studying secondary structures of biopolymers. The signals of *α*-helix, *β*-sheet, *β*-turn, and random coil structures, which are located precisely in the amide I, II, and III bands, can be provided [38]. Accurate analysis of the amide III region signal peak is a feasible way to quantify the secondary structure of proteins [39,40,41]. In this region, 1330–1295 cm^−1^ is designated as the region of the *α*-helix content, 1295–1270 cm^−1^ corresponds to the region of the *β*-turn content, 1270–1250 cm^−1^ and 1250–1220 cm^−1^ correspond to the regions of random coil and *β*-sheet content, respectively. Savitzky-Golay software is used to do 5-point smoothing and deconvolution. Appendix A shows the IR spectra of the samples of PGG (blank) and PGG in different mixed solvents system. Their secondary structures exhibited obvious differences. Quadratic polynomial integral calculation was carried out according to the literature reports. The statistics of secondary structure content are shown in Table 1.

Table 1 shows that the content of secondary structures in PGG aggregates was obviously affected by solvents. It is found that the sum of *α*-Helix + *β*-Sheet shows a certain regularity. In the protic solvent system, the sum of the *α*-Helix + *β*-Sheets tends to decrease with the reduction of polarity. In the aprotic solvent, the sums of *α*-Helix + *β*-Sheets are larger in THF and acetone systems. In addition, the Helix/Coil ratio shows the aggregation state of PGG in a select solvent. In protic solvents, the Helix/Coil ratios are higher in water, methanol, and ethanol systems, and the ratio tends to decrease with the decrease of polarity. In fact, stable emulsions were formed in water, methanol, and ethanol systems, and the solutions were stratified in 2-Propanol. The results indicate that the higher Helix/Coil ratios induce the stable aggregations in protic solvent. In aprotic solvent, the Helix/Coil ratios are higher in THF and acetone systems, and ordered gel phase was observed in acetone and THF systems with the evaporation of solvent for 2 h. Unexpectedly, The Helix/Coil of PGG-1,4-Dioxane sharply reduced, suggesting an additional phase transition into an unordered structure. The results indicate that hydrogen bonding ability dependent on the solvent polarity should play a significant role in improving interactions between components. In fact, stable emulsions were formed in water, methanol, and ethanol systems, whereas ordered gel phases were observed in acetone and THF systems with the evaporation of solvent for 2 h. These results suggest that the structure of *α*-Helix and *β*-Sheet can promote the formation of the ordered and stable aggregation structures.

The XRD patterns of all of the studied samples are listed in Figure 3. The peaks located in the region of 2θ of around 8° and 20° are associated with the diameter of the triple helix and the intensity of the reconstructed triple-helix structure of collagen [42]. With the addition of organic solvents, the peak at 2θ = 8° has changed, which interferes with the reassembling of the triple-helix structure of gelatin during the induction process. The diffraction around 20° were broad and there were no sharp peaks. The peak intensity was weakened as solvent changed from THF, to 1,4-dioxane, to acetone, to 2-propanol, to methanol, to ethanol, and to water, indicating decrease in crystallinity. The peak moved towards a smaller angle, indicating that the covalent structures in different solvents were different. There were also peaks at multiple angular positions (13–14°, 32°, 42°, etc.), indicating a weak phase separation within the system [43]. The solvent did not improve the non-crystalline properties of PGG, and there was a slight phase separation in the system. The solvent changed the conformation of PGG and increased the content of the ordered structure, but the disordered structure was still dominant.

### 2.3. Coating Performance

The coating performance of PGG was evaluated. Figure 4a shows that the friction resistance of PGG/acetone coating was the highest under both sweat and wet conditions. Acetone is a good solvent for PDMS. In acetone, PDMS should gather on the surface of aggregate. The size of the aggregate in acetone was smaller than that in ethanol, which was beneficial to the coating of aggregates on the surface of finished pigskin. The excellent flexibility and low *T*_g_ were conducive to enhancing the resistance to friction. Therefore, the friction resistance times of PGG/acetone coating were more than that of PGG/ethanol. Under both wet and sweat conditions, the friction resistances of these coatings were consistent.

Due to the similar compositions and structures of gelatin and collagen, PGG showed excellent adhesion to the surface of collagen–based materials. After UV irradiation for 2 h, the casein coating turned brownish-yellow, while the PGG coating had no obvious change in color (Figure 4c), which demonstrated that the PGG coating had excellent optical stability under UV radiation. Therefore, the PGG shows excellent coating performance for the pigskin substrate and has a good commercialization potential.

X-ray photoelectron spectroscopy (XPS) analysis can help to evaluate the performances of the coatings. The detailed XPS spectra of silicon in Figure 5 indicate several components. After coating, the Si 2p spectrum can be fitted into one peak centered at 101.5 eV. This peak can be assigned to silicon in Si–O bonds. In addition, the contents of silicon in the samples are obtained by XPS spectra, which are described in Table 2. Furthermore, PGG/acetone coating shows the highest silicon content, which helps to improve the friction resistance of the coating. This result is consistent with the results of the friction experiments.

### 2.4. Gel formation and Its Application

Figure 6 shows the SEM images of gel formed in acetone and THF. Gel formation can be driven by hydrophobic interactions, π-π stacking, and intermolecular hydrogen bonding (H-bonding) [44,45,46,47]. These interactions are inherent in the self-aggregation behavior of natural amino acids [45]. By utilizing these three driving forces, specific properties and functions of macromolecules can be obtained. For example, the convenient preparation of gels has been achieved through hydrophobic interactions [48]. A highly stable gel has been prepared by intermolecular hydrogen bonding between polymer chains and solvents [49]. The addition of solvent might lead to PGG aggregation and decreased solubility [50]. The addition of a polar solvent to the PGG solution increases the polarity of the solution and the interaction between the solvent and the reaction solution, thereby promoting the formation of the gel. The temperature and pH are constant during gel formation (room-temperature). In this study, PGG solution formed a physical gel in polar aprotic solvents by solvent-induced changes of hydrogen bonding, such as hydrogen bonding acceptor, which can break intramolecular hydrogen bonding, and induce intermolecular hydrogen bonding. Figure 6a–e and Figure 6f–j show that, as the PGG solution/acetone or THF (*v/v*) ratio decreased from 1:1 to 1:5, the solvent–PGG interactions, especially the hydrogen bonding, became strong enough to result in the aggregation of metastable gelator molecules and increase in gel orientation [43].

Figure 7 shows that the gel was still stable as the PGG solution/solvent ratio decreased to 1:10 (*v/v*). This phenomenon offers the possibility of using PGG for organic wastewater treatment in the current chemical industry, such as the pharmaceutical and catalysis industries. Gelatin and PDMS chains in PGG are non-toxic. They are widely used in the chemical industry. Therefore, PGG cannot cause secondary pollution.

## 3. Materials and Methods

### 3.1. Materials

Type A gelatin from pigskin was purchased from China National Pharmaceutical Group Corporation (Beijing, China) and used after dialysis. The isoelectric point (pI, 8.5) of the dialyzed gelatin after complete deionization was determined by fluorescence spectroscopy. Sodium dodecyl sulfate (SDS) was purchased from Alfa Aesar and recrystallized from ethanol before use. Ally glycidyl ether (AGE) and chloroplatinic acid hexahydrate (H_2_Pt_6_Cl_6_•6H_2_O) were obtained from Alfa Aesar. Hexamethylcyclotrisiloxane (D3, >95%), *n*–butyllithium (C_4_H_9_Li, >99%), and chlorodimethylsilane (C_2_H_7_ClSi, >99%) were purchased from Sigma–Aldrich (St. Louis, MO, USA). Benzene, deionized water (conductivity = 2.06 µS cm^−1^, dielectric constant ε = 80.40), methanol (ε = 32.70), ethanol (ε = 24.50), isopropanol alcohol (ε = 17.90), acetone (ε = 20.70), tetrahydrofuran (THF, ε = 7.58), and 1,4-dioxane (ε = 2.25) solvents (China National Pharmaceutical Group Corporation) were all of analytical reagent (AR) grade and strictly dehydrated before use.

### 3.2. Synthesis of α–[3–(2,3–epoxy–propoxy)propyl]–ω–butyl–polydimethylsiloxanes (PDMS-E) Grafted Gelatin

D3, C_4_H_9_Li, and C_2_H_7_ClSi were used to synthesize polydimethylsiloxanes with a Si–H group at one end (PDMS-H) through anionic addition polymerization. First, 10 mL of benzene were added to a flask and then 24 mL of C_4_H_9_Li were added. After reducing pressure and ventilation with argon gas, 45.99 g of D3 resolved in 40 mL of benzene was added to the flask. After reaction for 30 min, 50 mL of THF was added into the system to react for 8 h. Then, 11 mL of C_2_H_7_ClSi was injected into the flask to stop the reaction. The solution was first filtered by a sand-core filter to remove the lithium chloride precipitate. Then, the filtrate was distilled under reduced pressure at 50 °C (−0.01 Mpa) to remove the low-boiling-point solvent. Subsequently, the temperature was increased to 90 °C to remove the unreacted D3 to obtain purified PDMS-H (Appendix A). PDMS-E was then prepared by hydrolyzation of PDMS-H and AGE under Pt–catalyst (*M_w_* = 1.14 × 10^3^ g mol^−1^, *M_w_*/*M_n_* = 1.16, Appendix A).

All gelatin samples were prepared from a stock solution of dialyzed gelatin in order to minimize experimental errors. The 130-mL stock solution was prepared by dissolving gelatin in distilled water (5 wt%) and stirring for 3 h at 50 °C. Subsequently, the pH of the stock solution was adjusted to 10.0 using NaOH solution (2.0 mol L^−1^, about 220 µL). SDS was added to the gelatin solution with SDS concentration set at 0, 0.5, 1.0, 1.5, 2.0, 2.5, 3.0, and 3.5 g/L^−1^. The solution was stirred for 6 h. Then, PDMS-E was added to the above solution at 50 °C at a rate of 20 drops/min^−1^ with stirring until the epoxy groups or primary amino groups ratio reached 0.8:1.0 (mol:mol). The reaction was allowed to continue for another 24 h. After reaction for 24 h, the solutions were cooled to 5 °C for concentration, and the content of free −NH_2_ groups was tested by the Van Slyke method at 40 °C and trinitrobenzenesulfonic acid (TNBS) assay at pH = 10.5 [20]. Finally, the pH of the reaction solution was adjusted to 7.0 with HCl solution (2.0 mol L^−1^, about 150 µL).

### 3.3. Aggregation of PGG in Mixed Solvent

The 130 mL stock solution was divided into 13 equal parts. One part was used as blank, and organic solvents were added to the other parts. The organic solvents included methanol, ethanol, isopropanol alcohol, acetone, THF, and 1,4-dioxane. We calculated the amount of water in the reaction liquid, and based on this, we set the amount of solvent, adding different proportions of solvent. The volume ratios of reaction liquid to solvent were set to 1:1 or 1:2.

### 3.4. Characterization

High-resolution transmission electron microscopy (HR-TEM) images were acquired on a Tecnai JEM-2100 (Japan Electronics Co., Ltd., Tokyo, Japan) equipped with a charge coupled device (CCD) camera (Gatan Bioscan, Pleasanton, CA, USA) at 100 kV with a point resolution higher than 0.19 nm. Fourier transform infrared (FTIR) spectra were recorded using a Bruker Tensor–27 Fourier-transform infrared spectrometer (spectral range between 4000 and 450 cm^−1^, Bruker Co., Billerica, MA, USA). Solution samples were dried by freeze drying. Thirteen samples were obtained and were measured at room temperature in the solid state using a single reflection diamond attenuated total reflectance. X-Ray diffraction (XRD) patterns were characterized using a Bruker D8 advanced X-ray powder diffractometer with Cu-Ka radiation (λ = 1.5418 Å). Scanning electron microscopy (SEM) images were performed on FEI Quanta 200 with accelerating voltage 20 kV. The X-ray photoelectron spectroscopy (XPS) were carried out by a Thermo Fisher Scientific ESCALAB 250 spectrometer (ThermoScientific Co., Waltham, MA, USA) with a pass energy of 20 eV and a power of 60W (=5 mA × 12 kV) under the Al Kα line (1486.6 eV).

### 3.5. Coating Performance

First, a piece of pigskin was degreased after fleshing and then re-tanned with resin. The finished pigskin was cut into samples with size of 30 cm × 30 cm and sixteen samples were obtained. Second, 160 g of PGG solution was mixed into ethanol or acetone at Vsolvent:Vsolution = 2:1 (*v/v*). Solid content was about 15% in the mixed system. Casein paint sample was prepared by dissolving 1.5 g of 256# (trade name of casein paint, made in San Francisco; casein paint, derived from milk protein, is a fast-drying, water-soluble medium used by artists) into 100 g of water. Third, 30.0 g of mixture of PGG solution and ethanol or acetone was coated onto the surface of the finished pigskin. Additionally, 10.0 g of PGG solution and 10.0 g of 256# solution were directly coated onto the surface of finished pigskin, respectively. All coated pigskin samples were laid in an environment with the ideal temperature and humidity for three days and then tested for their resistance to friction under dry, wet, and sweat conditions. Uncoated finished pigskin was used for reference. The samples were placed in the friction machine for testing. The degree of surface wear was designed and friction times were compared for different coatings. In addition, the pigskins coated with 10.0 g of PGG solution and 10.0 g of 256# solution were compared in their resistance to radiation. The samples were laid under UV light (λ = 365 nm) for 2 h and the color variation was compared. Finally, the pigskin coated with 10.0 g of PGG solution was used to test the adhesion index of the coating layer.

## 4. Conclusions

In this paper, the aggregation of PGG in water was tuned by six organic solvents, including methanol, ethanol, 2-propanol, acetone, tetrahydrofuran, and 1,4-dioxane. TEM, IR, XRD, and SEM analyses showed that the aggregation pattern of PGG was affected by the solvent polarity and selectivity. PGG showed regular spherical morphology in polar solvents, including water, methanol, ethanol, 2-propanol, and acetone, and exhibited excellent coating performance on the surface of pigskin. Gel was formed in acetone and THF with the slow evaporation of solvent, and gel orientation increased with increase in the volume proportion of organic solvent. This property offers the possibility of applying PGG to organic sewage treatment in the chemical industry.

## Figures and Tables

**Figure 1 molecules-24-02264-f001:**
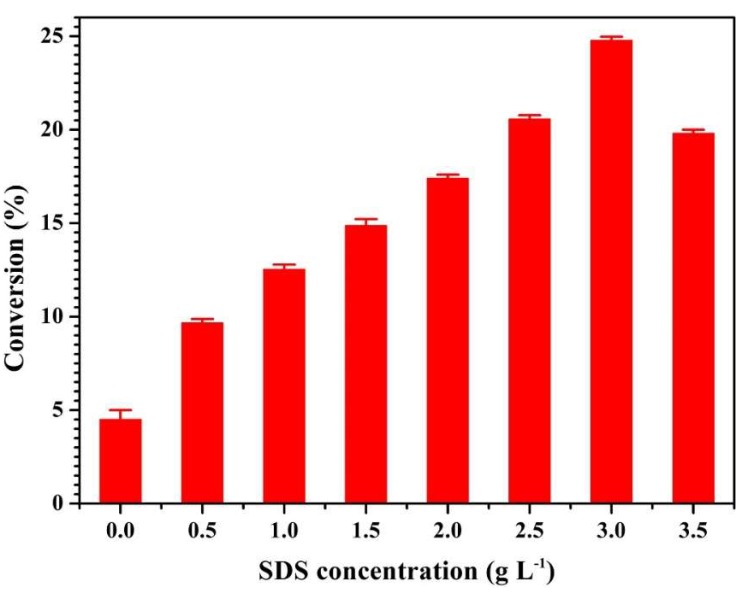
Conversion degree of free −NH_2_ groups at 5% (*w*/*w*) gelatin with the increasing of sodium dodecyl sulfate (SDS) concentration from 0 g L^−1^ to 3.5 g L^−1^. Data represent mean ± SD, *N* = 3.

**Figure 2 molecules-24-02264-f002:**
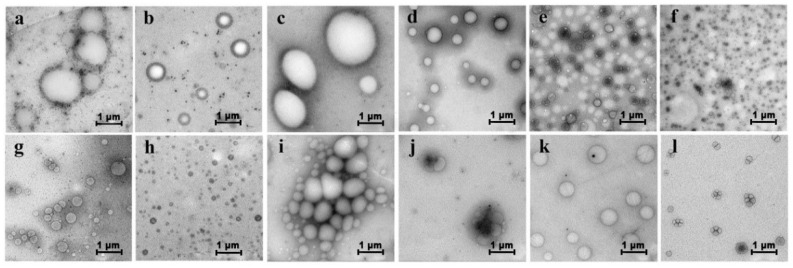
Morphologies of PGG when PGG solution was mixed with methanol (**a**,**g**), ethanol (**b**,**h**), isopropanol alcohol (**c**,**i**), acetone (**d**,**j**), THF (**e**,**k**), and 1,4-dioxane (**f**,**l**) at solution/solvent (*v/v*) ratios of 1:1 (**a**–**f**) and 1:2 (**g**–**l**).

**Figure 3 molecules-24-02264-f003:**
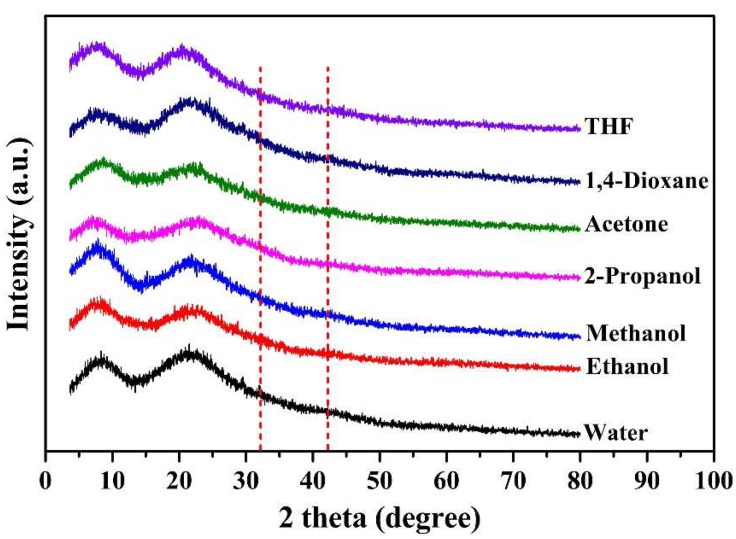
XRD patterns of PGG when PGG solution was mixed with six organic solvents (PGG solution/solvent *v/v* = 1:1).

**Figure 4 molecules-24-02264-f004:**
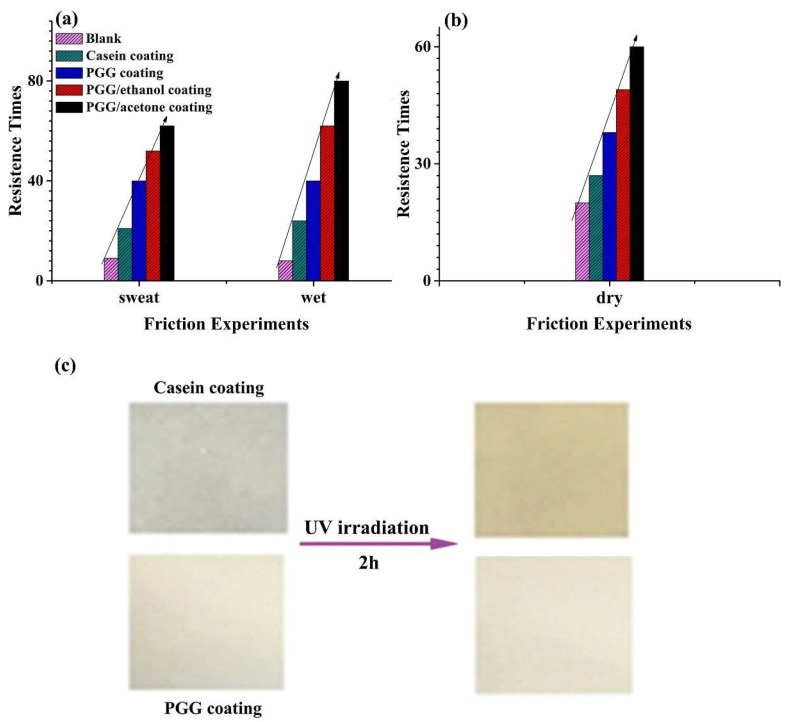
Friction resistance of pure pigskin (blank) and pigskin coated with PGG/acetone, PGG/ethanol, PGG, and casein, respectively, under wet and sweat conditions (**a**); friction experiment under dry conditions (**b**); casein and PGG coatings under UV radiation (**c**).

**Figure 5 molecules-24-02264-f005:**
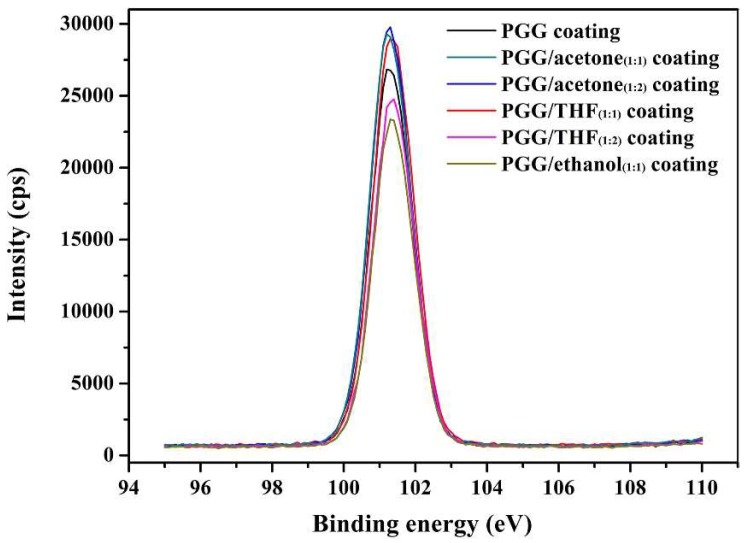
X-ray photoelectron spectroscopy of Si 2p profiles for different coatings.

**Figure 6 molecules-24-02264-f006:**
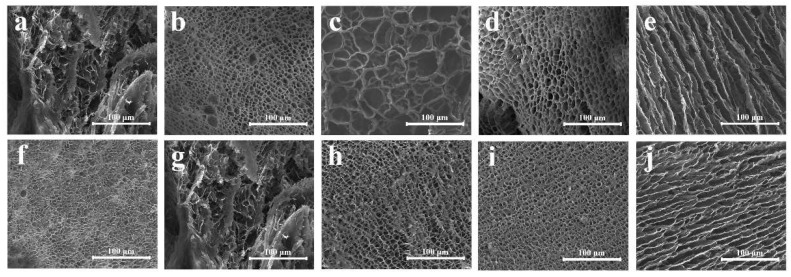
SEM images of gels formed in different solvents: (**a**–**e**) Acetone was the solvent and PGG solution/solvent (*v/v*) ratio changed from 1:1 to 1:5; (**f**–**j**) THF was the solvent and PGG solution/solvent (*v/v*) ratio changed from 1:1 to 1:5.

**Figure 7 molecules-24-02264-f007:**
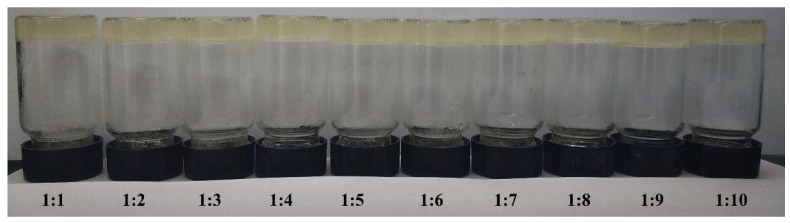
Photos of gels formed in THF; the ratio of THF to water is 1:1 to 1:10 (*v/v*). The volume of mixed solution in every part was 2 mL.

**Table 1 molecules-24-02264-t001:** Secondary structure content of epoxy-terminated polydimethylsiloxane (PDMS-E) grafted gelatin (PGG) after PGG solution was mixed with six organic solvents (PGG solution/solvent *v/v* = 1:1). Results are mean ± standard deviations of duplicate analysis. Values followed by different letters in the same line are significantly different (*p* ≤ 0.05, *N* = 3).

Solvents	Blank	Methanol	Ethanol	2-Propanol	Acetone	Tetrahydrofuran	1,4-Dioxane
*α*-Helix	24.45 ± 0.18 ^b^	27.99 ± 0.16 ^a^	28.07 ± 0.31 ^a^	14.59 ± 0.42 ^d^	17.73 ± 0.08 ^c^	24.48 ± 0.11 ^b^	12.55 ± 0.34 ^e^
*β*-Sheet	31.31 ± 0.07 ^b^	26.55 ± 0.20 ^d^	21.50 ± 0.03 ^f^	29.83 ± 0.17 ^c^	33.54 ± 0.36 ^a^	30.18 ± 0.23 ^c^	24.79 ± 0.16 ^e^
*β*-Turn	17.50 ± 0.12 ^c^	11.34 ± 0.28 ^d^	18.71 ± 0.15 ^a^	18.33 ± 0.10 ^ab^	18.50 ± 0.29 ^ab^	17.95 ± 0.13 ^bc^	18.65 ± 0.22 ^ab^
Random Coil	26.73 ± 0.26 ^f^	33.86 ± 0.40 ^c^	31.73 ± 0.22 ^d^	37.24 ± 0.06 ^b^	30.21 ± 0.15 ^e^	27.39 ± 0.17 ^f^	44.01 ± 0.31 ^a^
*α*-Helix+*β*-Sheet	55.76 ± 0.24 ^a^	54.54 ± 0.33 ^a^	49.57 ± 0.29 ^c^	44.42 ± 0.45 ^d^	51.27 ± 0.40 ^b^	54.66 ± 0.30 ^a^	37.34 ± 0.49 ^e^
Helix/Coil	0.91 ± 0.27 ^a^	0.83 ± 0.16 ^ab^	0.88 ± 0.06 ^a^	0.39 ± 0.17 ^ab^	0.59 ± 0.07 ^ab^	0.89 ± 0.18 ^a^	0.29 ± 0.02 ^b^

**Table 2 molecules-24-02264-t002:** XPS Elemental Analyses of different coatings. Results are mean ± standard deviations of duplicate analysis. Values followed by different letters in the same line are significantly different (*p* ≤ 0.05, *N* = 3).

Coating	Si 2p (%)
PGG coating	19.52 ± 0.19 ^b^
PGG/acetone_(1:1)_ coating	20.56 ± 0.28 ^a^
PGG/acetone_(1:2)_ coating	20.90 ± 0.22 ^a^
PGG/THF_(1:1)_ coating	20.34 ± 0.16 ^a^
PGG/THF_(1:2)_ coating	18.85 ± 0.21 ^b^
PGG/ethanol_(1:1)_ coating	18.04 ± 0.13 ^c^

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
