# Peer review of "Induced Aggregation of Epoxy Polysiloxane Grafted Gelatin by Organic Solvent and Green Application"

_molecules, 2019, doi:10.3390/molecules24122264_

Round 1

Reviewer 1 Report

The authors report aggregation of gelatin. There are some useful data for the possible application. However, there are some problems of the manuscript. One major issue is the discussion that is superficial. Comparison and discussion on the possible reasons/approaches need further enhancement.

Highlight 1: change ‘polarity’ to ‘polarities’.

Line 76-81: use past tense as it described what you did.

Line 85: ‘pI’ is widely used for the isoelectric point. Why did you use ‘IP’ here?

Line 209-210: label the positions 32° and 42° in this Figure as they were not obvious.

Lines 218-232: it is confusing of the paragraphs. The y axes of Fig. 43 are ‘retention times. However, Lines 229-230 described colur, how to understand the logic behind?’.

Fig. 1: What kind of information the readers can get from these Figures. For each group, diverse of the morphology pattern within a same group is very complicated thus what’s the rule we can get?

Lines 179-193: Amide A and fingerprint zone are more meaningful (for instance, Food Hydrocolloid, 94, 459-467), why did the author focus on amide I, II and III? Further discussion is needed.

Table 1: add SD for each data and statistical analysis among different groups should be conducted.

Section 3.3: What’s the approach for this aggregation of PGG? Need more intermediate steps to show the pathway under the current conditions (for instance, Food Chem, 277, 327-335). 

For gelatin aggregation, a very important parameter to evaluate proteins of the gelatin system is helix/coil ratio. The authors are suggested to discuss more on this point. Furthermore, what kind of possible approach to get the final aggregation including kinetics investigation.

Fig. 4: for the scale 100µm, there should be a space between the number and the unit.

Author Response

Response to Reviewer 1 Dear Sir or Madam: Attached please find our revised manuscript (molecules-517046) entitled “Induced Aggregation of Epoxy Polysiloxane Grafted Gelatin by Organic Solvent and Green Application” submitted to the journal molecules. Thank you very much for your kind reading of our manuscript and helpful comments. Now, the manuscript is revised under the guidance of the comments. The details of how we revised our manuscript and the response to the comments are given as follows: (1) Comment Highlight 1: change ‘polarity’ to ‘polarities’. Response  According to your comment, the ‘polarity’ has been replaced by to ‘polarities’, Lines 17. (2) Comment Line 76-81: use past tense as it described what you did. Response  According to your comment, these incorrect sentences have been revised. Lines 79, ‘are’ has been replaced by to ‘were’; Lines 81, ‘present’ has been replaced by to ‘presented’; Lines 82 ‘is’ has been replaced by to ‘was’. (3) Comment Line 85: ‘pI’ is widely used for the isoelectric point. Why did you use ‘IP’ here? Response  I am very sorry for the mistake. Thanks for your reminding. Lines 87, ‘IP’ has been corrected for ‘pI’. (4) Comment Line 209-210: label the positions 32° and 42° in this Figure as they were not obvious. Response  According to your comment, the Figure 3 has been recreated, Lines 237-238. (5) Comment Lines 218-232: it is confusing of the paragraphs. The y axes of Fig. 4 are ‘retention times. However, Lines 229-230 described colur, how to understand the logic behind?’. Response  I'm sorry for the confusion caused by my unclear expression. The Figure 4 has been revised. We believe the logic is better now. The coating performance of PGG was evaluated. Figure 4a shows that the friction resistance of PGG/acetone coating was the highest under both sweat and wet conditions. Acetone is a good solvent for PDMS. In acetone, PDMS should tend to gather on the surface of aggregate. The size of aggregate in acetone was smaller than that in ethanol, which was beneficial to the coating of aggregates on the surface of finished pigskin. The excellent flexibility and low Tg were conducive to enhancing the resistance to friction. Therefore, the friction resistance times of PGG/acetone coating was more than that of PGG/ethanol. Under both wet and sweat condition, the friction resistances of these coatings were consistent. Due to the similar compositions and structures of gelatin and collagen, PGG showed excellent adhesion to the surface of collagen–based materials. After UV irradiation for 2 h, the casein coating turned brownish yellow, while the PGG coating had no obvious change in color (Figure 4c), which demonstrated that the PGG coating had excellent optical stability under UV radiation. Therefore, the PGG shows excellent coating performance for the pigskin substrate and has a good commercialization potential. (6) Comment Fig. 1: What kind of information the readers can get from these Figures. For each group, diverse of the morphology pattern within a same group is very complicated thus what’s the rule we can get? Response  I am sorry that my expression is unclear. The explanation has been revised in the revised manuscript. Line 181-198, Figure 2 shows that PGG presented different morphologies in different solvents. Spherical aggregates were formed in methanol, ethanol, isopropanol alcohol and acetone system at PGG solution/solvent ratio of 1:1 (v/v, Figure 2a, b, c and d), and smaller-scale spherical aggregates were observed as PGG solution/solvent ratio decreased to 1:2 (v/v, Figure 2g, h, i). But in acetone system, the decreasing of PGG solution/acetone ratio resulted in the formation of coacervate (Figure 2d vs. j). Interestingly, double-layer structure was formed in THF-water system at PGG solution/solvent ratio of 1:1 (v/v) (Figure 2e). The structure evolved to spherical aggregate at PGG solution/THF ratio of 1:2 (v/v, Figure 2k). Disordered aggregates were observed in 1,4-dioxane system (Figure 2f and l). These results suggest that the morphology transformations of PGG aggregates closely depend on the solvent polarity and the hydrogen bonding ability. In polar protic solvent including methanol, ethanol and isopropanol alcohol, spherical aggregates tend to form. With the increasing in the ratio of solvent, the scale of aggregate is decreasing. However, in nonpolar aprotic solvent THF and 1,4-dioxane, complex morphologies can be obtained, such as double-layer, disordered aggregates or coacervate. (7) Comment Lines 179-193: Amide A and fingerprint zone are more meaningful (for instance, Food Hydrocolloid, 94, 459-467), why did the author focus on amide I, II and III? Further discussion is needed. Response  Thank you for your suggestion. According to your comment, the literature was added to the revised manuscript. Further, discussions were supplemented, Lines 202-204. (8) Comment Table 1: add SD for each data and statistical analysis among different groups should be conducted. Response  According to your comment, The SD has been added in Table 1, Lines 216-218. (9) Comment Section 3.3: What’s the approach for this aggregation of PGG? Need more intermediate steps to show the pathway under the current conditions (for instance, Food Chem, 277, 327-335). Response  Thank you for your suggestion. According to your comment, the literature was added to the revised manuscript. Further, discussions were supplemented, Lines 286-289. (10) Comment For gelatin aggregation, a very important parameter to evaluate proteins of the gelatin system is helix/coil ratio. The authors are suggested to discuss more on this point. Furthermore, what kind of possible approach to get the final aggregation including kinetics investigation. Response  According to your comment, the Helix/Coil ratio was added in Table 1. The Helix/Coil ratio shows the aggregation state of PGG in a select solvent. In protic solvents, the Helix/Coil ratios are higher in water, methanol, ethanol systems, and the ratio tends to decrease with the decrease of polarity. In fact, stable emulsion was formed in water, methanol and ethanol systems, and the solution is stratified in 2-Propanol. The results indicate that the higher Helix/Coil ratios induce the stable aggregations in protic solvent. In aprotic solvent, the Helix/Coil ratios are higher in THF and acetone systems, and ordered gel phase was observed in acetone and THF systems with the evaporation of solvent for 2 h. Unexpectedly, The Helix/ Coil of PGG-1,4-Dioxane sharply reduced suggesting an additional phase transition into an unordered structure. The kinetics investigation is a very good suggestion, and we will definitely use the kinetic method to discuss the aggregation approach in future experiments. However, the experiment has not been supplemented now. Because the room temperature is higher, the sol-gel properties are affected by the temperature. The room temperature has increased by more than 10 degrees compared with that at that time, so the error of repeated experiments in the current experimental environment may be very large. (11) Comment Fig. 4: for the scale 100µm, there should be a space between the number and the unit. Response  According to your comment, the Figure 6 has been revised. Thank you very much for your good job again. Any question/information please corresponds to the CORRESPONDING AUTHOR Jing Xu.                                        Sincerely                                        Jing Xu                                        E-mail address: xujing@qlu.edu.cn

Reviewer 2 Report

This manuscript reported the aggregation of amphiphilic polymer PDMS-E grafted gelatin (PGG) in water induced by methanol, ethanol, 2-propanol, acetone, tetrahydrofuran (THF) and 1,4-dioxane. The PGG aggregates had regular spherical morphology in polar solvents and exhibited excellent coating performance on the surface of pigskin. In addition, gel was formed in acetone and THF which offers the possibility of applying PGG to organic sewage treatment in chemical industry field. Generally speaking, I suggest that this article can be published after minor revision. My detailed comments are as follow:

(1) The author studied the aggregation of amphiphilic polymer PDMS-E grafted gelatin and the grafting rate is very important. Therefore, the experiment to determine the graft ratio is suggested.

(2) In page3,line 120,the author said “The volume ratios of reaction liquid to solvent were set to 1:1 or 1:2.” However, at these two ratios, the concentration of PGG will change. Are these changes affect the aggregation of PGG?

(3) In page 4,line 163,the author only simply described the phenomenon “Similarly, with the PGG solution/solvent ratio decreasing, the scale of aggregates decreased in ethanol and isopropanol alcohol” Maybe some explanation is needed here.

(4) In “coating performance” part, some necessary characterizations are suggested, such as X-ray photoelectron spectroscopy and Scanning electron microscopy.

(5) In page 7,line 245,the author demonstrated “In this study, PGG solution formed a physical gel in polar aprotic solvents by solvent-induced change of hydrogen bonding” Does the gel dissolve when it breaks hydrogen bonds (such as heating or changing pH)?

(6)In figure 5, inverted cups may be more reflective of the state of the gel.

(7) There are some non-uniform formats in the article, such as in figure 1, the scale bar of b and h is obviously much longer than others.

(8) The authors could add the following references which would again increase the interest to general Polysiloxane material readers:  Polymer, 2019, 162, 58-62 ; Polymer, 2017, 125, 303-329.

Author Response

Response to Reviewer 2 Dear Sir or Madam: Attached please find our revised manuscript (molecules-517046) entitled “Induced Aggregation of Epoxy Polysiloxane Grafted Gelatin by Organic Solvent and Green Application” submitted to the journal molecules. Thank you very much for your kind reading of our manuscript and helpful comments. Now, the manuscript is revised under the guidance of the comments. The details of how we revised our manuscript and the response to the comments are given as follows: (1) Comment The author studied the aggregation of amphiphilic polymer PDMS-E grafted gelatin and the grafting rate is very important. Therefore, the experiment to determine the graft ratio is suggested. Response  Thank you for your suggestion. According to your comment, the grafting rate was supplemented in Figure 1, lines 159-170. The detail of experiment was also supplemented, lines 117-119. (2) Comment In page3, line 120, the author said “The volume ratios of reaction liquid to solvent were set to 1:1 or 1:2.” However, at these two ratios, the concentration of PGG will change. Are these changes affect the aggregation of PGG? Response  Thank you for your suggestion. The study discussed the influence of the solvent ratio on PGG aggregation. Firstly, the volume of water was calculated from the volume of reaction solution based on the material mass percentage. Then, the volumes of solvent can be determined. At these two ratios, the effect of aggregation of PGG can be reflected. (3) Comment In page 4, line 163, the author only simply described the phenomenon “Similarly, with the PGG solution/solvent ratio decreasing, the scale of aggregates decreased in ethanol and isopropanol alcohol” Maybe some explanation is needed here. Response  I am sorry that my expression is unclear. The explanation has been revised in the revised manuscript. Figure 2 shows that PGG presented different morphologies in different solvents. Spherical aggregates were formed in methanol, ethanol, isopropanol alcohol and acetone system at PGG solution/solvent ratio of 1:1 (v/v, Figure 2a, b, c and d), and smaller-scale spherical aggregates were observed as PGG solution/solvent ratio decreased to 1:2 (v/v, Figure 2g, h, i). But in acetone system, the decreasing of PGG solution/acetone ratio resulted in the formation of coacervate (Figure 2d vs. j). Interestingly, double-layer structure was formed in THF-water system at PGG solution/solvent ratio of 1:1 (v/v) (Figure 2e). The structure evolved to spherical aggregate at PGG solution/THF ratio of 1:2 (v/v, Figure 2k). Disordered aggregates were observed in 1,4-dioxane system (Figure 2f and l). These results suggest that the morphology transformations of PGG aggregates closely depend on the solvent polarity and the hydrogen bonding ability. In polar protic solvent including methanol, ethanol and isopropanol alcohol, spherical aggregates tend to form. With the increasing in the ratio of solvent, the scale of aggregate is decreasing. However, in nonpolar aprotic solvent THF and 1,4-dioxane, complex morphologies can be obtained, such as double-layer, disordered aggregates or coacervate. Revised manuscript, In page 5, lines 181-194. (4) Comment In “coating performance” part, some necessary characterizations are suggested, such as X-ray photoelectron spectroscopy and Scanning electron microscopy. Response  Thank you for your suggestion. According to your comment, XPS data and discussion were supplemented in revised manuscript, in Figure 5 and Table 2. In page 9. (5) Comment In page 7, line 245, the author demonstrated “In this study, PGG solution formed a physical gel in polar aprotic solvents by solvent-induced change of hydrogen bonding” Does the gel dissolve when it breaks hydrogen bonds (such as heating or changing pH)? Response According to your comment, the explanation has been revised in the revised manuscript. The temperature and pH are constant during gel formation (room-temperature). In page 9, lines 289-290. (6) Comment In figure 5, inverted cups may be more reflective of the state of the gel. Response  Thank you for your suggestion. According to your comment, the Figure 7 in the revised manuscript has been revised. (7) Comment There are some non-uniform formats in the article, such as in figure 1, the scale bar of b and h is obviously much longer than others. Response  According to your comment, the Figure 2 in the revised manuscript has been revised. (8) Comment The authors could add the following references which would again increase the interest to general Polysiloxane material readers:  Polymer, 2019, 162, 58-62; Polymer, 2017, 125, 303-329. Response  Thank you for your suggestion. According to your comment, these literatures were added to the revised manuscript and cited in the discussions. Thank you very much for your good job again. Any question/information please corresponds to the CORRESPONDING AUTHOR Jing Xu.                                        Sincerely                                        Jing Xu                                        E-mail address: xujing@qlu.edu.cn

Round 2

Reviewer 1 Report

The manuscript has been improved significantly.

Author Response

Response to Editor

Dear Academic Editor:

We have revised the manuscript according to your comments. We look forward to hearing from you. Thank you very much for your kind reading of our manuscript and helpful comments. Now, the manuscript is revised under the guidance of the comments. All changes indicate in red color in the revised manuscript. The details of how we revised our manuscript and the response to the comments are given as follows:

(1) Comment

Row 118: a reference for the method used must be added

Response  Thank you for your suggestion. According to your comment, the literature was added to the revised manuscript. Row 113.

(2) Comment

Caption of Figure 1: on my opinion “conversion degree” should be better than the term “rate”. Please check the text that refers to Figure 1.

Response  According to your comment, the “conversion rate” has been replaced by to “conversion degree”, Lines 155, 157 and 163.

(3) Comment

Figure 3: XRD patterns. Why did the Authors only show XRD patterns starting from 10°/2 theta? It is known that gelatin has two main reflections: one centered at about 8°, related to the diameter of the triple helix, and a broad band at around 21°/2 theta related to the distance between amino acidic residues along the helix. The presence of the band at 8° is indicative of the triple helix, so it should be displayed and discussed.

Response  Thank you for your suggestion. According to your comment, the Figure 3 has been revised. The explanation has been revised in the revised manuscript. Lines 230-234, the XRD patterns of all of the studied samples are listed in Figure 3. The peaks located in the region of 2θ of around 8° and 20° are associated with the diameter of the triple helix and the intensity of the reconstructed triple-helix structure of collagen [44]. With the addition of organic solvents, the peak at 2θ = 8° has changed, which interferes with the reassembling of the triple-helix structure of gelatin during the induction process.

Thank you very much for your good job again. Any question/information please corresponds to the CORRESPONDING AUTHOR Jing Xu.

                                        Sincerely

                                        Jing Xu

                                        E-mail address: xujing@qlu.edu.cn
